# Sexual Dysfunction in Multiple Sclerosis: The Role of Executive Function

**DOI:** 10.3390/bs13050369

**Published:** 2023-04-29

**Authors:** Silvia Marinetto, Alice Riccardi, Filippo Barbadoro, Veronica Pucci, Enrico Selini, Aghite Pavan, Francesca Rinaldi, Paola Perini, Marco Puthenparampil, Paolo Gallo, Sara Mondini

**Affiliations:** 1Department of Philosophy, Sociology, Education and Applied Psychology, University of Padua, 35133 Padua, Italy; silvia.marinetto@studenti.unipd.it (S.M.); veronica.pucci@phd.unipd.it (V.P.);; 2Multiple Sclerosis Centre, Department of Neuroscience, University-Hospital of Padua, 35128 Padua, Italy; alicericcardi13@gmail.com (A.R.); marco.puthenparampil@unipd.it (M.P.); paolo.gallo@unipd.it (P.G.); 3Human Inspired Technology Centre (HIT), University of Padua, 35121 Padua, Italy; 4Multiple Sclerosis Centre, University-Hospital of Padua, 35128 Padua, Italydr.paola.perini@gmail.com (P.P.)

**Keywords:** Multiple Sclerosis, sexual dysfunction, cognitive abilities, executive functions, depression, self-perception

## Abstract

Cognitive impairment and sexual dysfunction are common symptoms in persons with Multiple Sclerosis (MS). The present study focuses on the relationship between these two dimensions by means of a specific assessment commonly used in clinical practice with this population. Fifty-five persons with a diagnosis of MS underwent specific cognitive tests and answered clinical questionnaires. Two cognitive tests, one for memory (the Selective Reminding Test), and one for attention (the Symbol Digit Modalities Test), were administered together with two tests for executive functions (the D-KEFS Sorting Test and Stroop Test). Two self-report questionnaires to investigate clinical, psychological and sexual features (the Beck Depression Inventory-II and Self-perception of Cognition in Multiple Sclerosis and Multiple Sclerosis Intimacy and Sexuality Questionnaire-19), were also administered. The main result highlights that sexual difficulties are associated with cognitive deficits, particularly with executive disorders, but not with memory and attention. Furthermore, sexual difficulties are better explained when depression symptoms are also taken into account. This study disentangles the interaction between sexual dysfunction, cognitive impairment and depression in persons with MS by emphasising the role of very high cognitive processing (i.e., executive functioning) in determining human behaviour.

## 1. Introduction

Multiple Sclerosis (MS) is a chronic neurological condition characterised by a wide range of symptoms that can severely affect the quality of life [1,2]. This inflammatory, chronic, and progressive pathology affects the Central Nervous System and is characterised by demyelination.

The clinical onset typically occurs between 20 and 40 years of age and is one of the most common causes of neurological disability in young adults [3]. Depending on the neural areas involved, the symptomatology is heterogeneous and could manifest as optic neuritis, hemiparesis, urinary and/or faecal sphincter dysfunctions, tremors, etc. [4,5]. Cognitive deficits, psychiatric symptoms and sexual dysfunction are also frequent manifestations.

A percentage of around 40–65% of people with MS experience cognitive decline [6,7], with the most affected domains being memory, attention, information processing and executive functions [8]. Even with minimal physical disability, cognitive deficits may strongly affect patients’ quality of life, for example, by interfering with their occupation. As reported by Chiaravalloti and DeLuca [6], 40–80% of persons with MS are unemployed because of cognitive difficulties. Cognitive deficits also affect the social, sexual and emotional dimensions of these persons [9]. Considering the psychiatric symptoms, depression is present in about 50% of persons with MS, with a prevalence of two or three times higher than in the general population, and many factors, both biological and psychosocial, may contribute to depression [10]. Moreover, Atlantis and Sullivan’s review [11] highlights a bidirectional association between depression and sexual dysfunction (SD), a symptom which is present in about 40–80% of persons with MS [12]. Although the exact cause of SD is still debated [13], Foley and Iverson’s [14] model operationalises this disturbance in MS, according to the origin, in three levels: *primary, secondary,* and *tertiary. Primary* sexual dysfunction stems directly from the demyelination of brain or spinal cord areas directly involved in genital control. It could manifest itself as decreased libido, genital numbness, difficulty in arousal and reaching orgasm, reduced vaginal lubrication or erectile/ejaculatory dysfunction [15]. *Secondary* sexual dysfunction refers to the indirect, non-sexual symptoms that nevertheless influence sexual response, such as pain, fatigue, weakness, spasticity, or poor cognitive functioning [15,16]. Lastly, *tertiary* sexual dysfunction can be defined as the psychological, social and cultural aspects deriving from a chronic disease that can indirectly interfere with sex life. Depression, anxiety, low self-esteem and change in social roles are some examples [16], as highlighted by Carotenuto et al. [17].

Sexual dysfunction is indeed a real issue in this pathology, as it is a symptom more frequently reported by persons with MS than people with other chronic disorders, such as rheumatoid arthritis, systemic lupus erythematosus, psoriatic arthritis, and ankylosing spondylitis, or than healthy people [18]. As mentioned above, cognitive impairment may be considered one of the non-genital symptoms of MS which contributes to sexual dysfunction. As a matter of fact, some studies investigating the correlates of sexual dysfunction in persons with MS have confirmed the presence of cognitive impairment [19,20,21]. For example, Demirkiran et al. [20] reported that problems in memory and concentration were significantly more common in persons with MS complaining about sexual dysfunction than those not reporting SD. Interestingly, Tepavcevic and colleagues [21] found a significant correlation between sexual and cognitive functioning in females but not in males with MS. However, these investigations administered cognitive tools that were not specific for people with MS, such as the MMSE (see Beatty and Goodkin [22] for a critical dissertation), which is specific for patients with dementia. In addition, these studies did not operationalise or measure the cognitive abilities selectively involved. Thus, our aim was to explore the relationship between sexual dysfunction, cognitive impairment and depression in people with MS. Using neuropsychological tests and self-report questionnaires targeting the population with MS, the specific aim was to identify which cognitive function is mainly involved in sexual difficulties.

## 2. Materials and Methods

### 2.1. Participants

Fifty-five persons aged between 19 and 62 (M = 39.76; SD = 11.25) from the Multiple Sclerosis Centre of the University Hospital of Padua (Italy) were enrolled in this study. Fifty-four of them were affected by Relapsing Remitting Multiple Sclerosis (RRMS), and only one by Primary Progressive Multiple Sclerosis (PPMS) [23,24]. Due to the fact that the difference among clinical subtypes of MS was not relevant for our aim, we decided to include also the single PPMS case. The group was composed of 38 females and 17 males, with education ranging from 5 to 19 years (M = 13; SD = 3.59). For each patient, clinical data on physical disability and on disease duration were collected. Physical disability was assessed by the Expanded Disability Status Scale (EDSS) [25], a clinical scale ranging from 0 (absence of disability) to 10 (death caused by MS), which evaluates neurological impairment.

### 2.2. Materials

During an in-person neuropsychological examination, all patients were administered the series of tools planned for the present investigation: three self-report questionnaires evaluating the presence of depression, self-perception of cognitive disorders and self-perception of sexual dysfunctioning. Subsequently, specific tests were administered to evaluate memory, attention and executive functions, which are the cognitive abilities most frequently impaired in Multiple Sclerosis [8].

### 2.3. Self-Report Questionnaires

This part of the evaluation consists of the self-administration of the Italian versions of the Multiple Sclerosis Intimacy and Sexuality Questionnaire-19 (MSISQ-19 [26]; Italian version in Carotenuto et al. [17]), the Self-Perception of Cognition in Multiple Sclerosis (Sclerosi Multipla Autopercezione Cognitiva, SMAC) [27] and the Beck Depression Inventory-II (BDI-II [28]; Italian version in Ghisi et al. [29]) to detect the clinical conditions and self-perception of participants with MS.

### 2.4. MSISQ-19 [17]

The MSISQ-19 is a 19-item questionnaire specifically designed to investigate self-perception of sexual dysfunctioning in the population with MS. The instrument has good psychometric properties and includes three subscales following Foley and Iversons’ classification of sexual dysfunction as *primary, secondary* and *tertiary*. Possible scores range from 19 to 95, with higher scores indicating greater sexual difficulties.

### 2.5. SMAC [26]

The SMAC is a 25-item self-report questionnaire aimed at investigating MS patients’ perceptions of their cognitive difficulties and requires patients to rate severity on a 5-point Likert scale. SMAC scores range from 0 to 100 and higher scores indicate greater severity of cognitive symptoms. This instrument was chosen to compare patients’ self-perception of both sexual and cognitive difficulties.

### 2.6. BDI-II [28]

Finally, the BDI-II was used to measure patients’ mood. According to the literature, depression is indeed a very common symptom in the MS population and it interferes with both cognitive [6] and sexual [11] functioning. Scores range from 0 to 63, with higher scores indicating greater depressive symptoms.

### 2.7. Cognitive Assessment

This part was designed to test specific cognitive abilities using appropriate tools for:

*Memory* (Selective Reminding Test, SRT from BRB-NT battery [30]; Italian version and Italian normative values in Amato et al. [31]);

*Attention* (Symbol Digit Modalities Test, SDMT from BRB-NT battery [30]; Italian version and Italian normative values in Amato et al. [31]);

*Executive functions* testing *Reasoning* (Delis–Kaplan Executive Function System Sorting Test, D-KEFS ST; [32] Italian version and Italian normative values in Mattioli et al., [33]; see the use of D-KEFS on the population with MS in Riccardi et al. [34]), and *Inhibition* (Stroop Test [34]; Italian version and Italian normative values in Amato et al. [30]).

### 2.8. Memory

The Selective Reminding Test (SRT), from Rao’s Brief Repeatable Battery of Neuropsychological Tests (BRB-NT) [31], was administered to evaluate different aspects of memory. This test gives three scores on the various phases of memory processing: Long-Term Storage (LTS) and Consistent Long-Term Retrieval (CLTR) for the learning phase, and the Selective Reminding Test–Delay (SRT-D) for the delayed retrieval. Possible scores for both LTS and CLTR range from 0 to 72, while possible scores for SRT-D range from 0 to 12. Lower scores indicate greater difficulties with memory.

### 2.9. Attention

The Symbol Digit Modalities Test (SDMT) [31] was selected to assess attention. The score gives information about sustained attention, visual tracking and processing speed, and is based on the number of correct answers given in 90 s (range between 0 and 110). A lower score indicates lower attentional abilities.

### 2.10. Executive Functions

Two different tests for executive functions were selected to evaluate different abilities of this complex cognitive domain: *Reasoning* using the Delis–Kaplan Executive Function System Sorting Test (D-KEFS ST) [33] and *Inhibition* using the Stroop Test [30]. The D-KEFS ST [33] investigates executive functions such as reasoning, categorisation, flexibility of thinking, problem-solving, concept-formation skills and abstraction. It is divided into two parts: Free Sorting (Free Sort Categorization, scores ranging from 0 to 16, and Free Sort Description, scores ranging from 0 to 64) and Sort Recognition (scores ranging from 0 to 64). Lower scores indicate poorer cognitive performance. The Stroop Test [35] investigates the ability to maintain cognitive control by inhibiting the automatic response triggered by stimuli. The score corresponds to the time (in seconds) taken to read 100 stimuli. Higher scores on this test indicate poor cognitive control and inhibition.

### 2.11. Statistical Analyses

Data were analysed with *RStudio* [36] considering demographical data (age and education), clinical features (EDSS and disease duration), scores on the neuropsychological tests (memory test, SRT; attention test, SDMT; executive functions tests, i.e., D-KEFS ST and STROOP), self-report questionnaires (SMAC and BDI-II) and measure of sexual dysfunction (MSISQ-19 global score and the scores on the three subscales on primary, secondary, and tertiary SD).

Correlations among variables were analysed through Pearson’s *r* to explore which variables co-occurred with SD. Furthermore, to better define the role of variables in predicting sexual dysfunctions, linear regression models were built with the MSISQ-19 global score as a dependent variable, and demographical data, clinical features, and scores on the neuropsychological tests as independent variables.

## 3. Results

In our sample, EDSS score ranged between 0 and 6.5 (M = 2.06; SD = 1.45). Disease duration estimated from the onset of the first symptoms ranged between 1 and 47 years (M = 11.2; SD = 9.24).

All descriptive statistics (whole sample and separately for males and females) results obtained from the questionnaires and the cognitive tests are reported in Table 1. Participants obtained a mean score of 35.13 (SD = 15.06) on the MSISQ-19, and an independent sample T-test showed no difference between males and females both in total scores (*t*(53) = 1.28, *p* = 0.202) and subscales (primary: *t*(53) = 1.677, *p* = 0.099; secondary: *t*(53) *= 1.139*, *p* = 0.259; tertiary: *t*(53) *= 0.79*, *p* = 0.432). Cronbach’s alpha confirmed MSISQ-19 as a reliable tool: the coefficient was 0.946 for the total 19-item scale, 0.877 for the primary subscale, 0.868 for the secondary subscale and 0.90 for the tertiary subscale.

No significant correlation was found between MSISQ-19 and demographical data. The MSISQ-19 total score and score of the three sub-scales did not correlate with disease duration, while only the secondary sub-scale significantly correlated with EDSS (*r* = 0.287, *p* = 0.03). As regards neuropsychological data, significant correlations were observed between the MSISQ-19 and D-KEFS ST and the Stroop test. In particular, the MSISQ-19 total score and those of the three sub-scales correlated with low performance on the D-KEFS Free Sorting Categorization (MSISQ-19 total: *r* = −0.481, *p* < 0.001 primary: *r* = −0.361, *p* = 0.007; secondary: *r* = −0.545, *p* < 0.001; tertiary: *r* = −0.400, *p* = 0.002) and on the D-KEFS Free Sorting Description (MSISQ-19 total: *r* = −0.409, *p* = 0.002; primary: *r* = −0.277, *p* = 0.041; secondary: *r* = −0.5, *p* < 0.001; tertiary: *r* = −0.324, *p* = 0.016 with the tertiary). The MSISQ-19 total score and the secondary sub-scale score correlated with low performance on the D-KEFS Sort Recognition (MSISQ-19 total: *r* = −0.311, *p* = 0.021; secondary: *r* = −0.371, *p* = 0.005). A significant correlation was also found between the secondary sub-scale and the Stroop Test (*r* = 0.349, *p* = 0.009). Finally, MSISQ-19 correlated both with depression, i.e., BDI-II (MSISQ-19 total: *r* = 0.746, *p* < 0.001; primary: *r* = 0.707, *p* < 0.001; secondary: *r* = 0.698, *p* < 0.001; tertiary: *r* = 0.660, *p* < 0.001) and self-perception of cognitive deficits, i.e., SMAC (MSISQ-19 total: *r* = 0.596, *p* < 0.001; primary: *r* = 0.610, *p* < 0.001; secondary: *r* = 0.499, *p* < 0.001; tertiary: *r* = 0.556, *p* < 0.001).

Furthermore, a series of linear regression models were carried out with MSISQ-19 total score as the dependent variable, and clinical and demographical data, neuropsychological tests and questionnaires as independent variables (Table 2). Clinical and demographical features never proved significant within the models, while D-KEFS ST was the only significant neuropsychological test within the models. The model fit was found to be improved by adding BDI-II or SMAC as covariates.

## 4. Discussion

Although some authors (e.g., [14,37,38]) have alluded to a relationship between cognitive decline and sexual dysfunction in persons with MS, the characteristics of this interaction have rarely been investigated (see also Pöttgen et al., [39]). Furthermore, the few existing studies adopted cognitive tools not suitable for this population [22] and reported ambiguous and contradictory results [19,20,21].

The purpose of our investigation was to focus on the association between cognitive decline and sexual dysfunction in people with MS by choosing assessment instruments widely employed with this specific population and taking into account the main factors at play (i.e., personal, clinical, psychological and cognitive data). In our sample, males and females equally experienced sexual difficulties, contrary to what was reported by Çelik et al. [40] and Zivadinov et al. [19], but in line with Demirkiran et al.’s [20] findings.

Physical disabilities measured with EDSS did not correlate, nor were they found to be predictive in the regression models with the global score of MSISQ-19, but EDSS correlated with the secondary scale of MSISQ-19, confirming that the presence of physical and functional difficulties may interfere with the sex life of patients. These results are consistent with Carotenuto et al. [17]. Although other authors [16] report that having a physical disability can be associated with low self-esteem and an altered self-image, our findings did not support the correlation between EDSS and the tertiary scale; this is in contrast with Carotenuto et al. [17]. It could be that the low–medium EDSS scores of most of our sample indicate mild physical difficulties that do not significantly interfere with self-esteem or perceived self-image.

Concerning cognitive performance, an interesting result is the relationship between MSISQ-19 scores and D-KEFS ST and the Stroop Test. Both neuropsychological tests measure several components of executive functions, such as categorisation skills, conceptual flexibility, problem-solving, abstract reasoning abilities and inhibitory control [30,31,32], whose decline seems to interfere with the sex life of persons with MS. In particular, impairment in every sub-component of D-KEFS ST seems to be predictive of sexual difficulties, while the *inhibitory* domain tested with the Stroop Test was only correlated with the secondary sub-scale of MSISQ-19. Interestingly, our findings showed that reasoning and categorisation measured on the D-KEFS Free Sorting component correlated with primary, secondary and tertiary sub-scales; however, according to Foley and Iverson’s [14] model, we were expecting correlations only with the secondary sub-scale. From a neuronal point of view, Fletcher et al. [36] highlighted that sexual functioning partially depends on higher cortical processes, and thus we may explain the correlation we found between D-KEFS Free Sorting and the primary sub-scale by assuming that higher cognitive processes such as executive functions may share the same cortical networks as sexual functions.

No relationship between memory, attention and self-reported sexual dysfunction was found in our sample, as was reported in previous studies which did not operationalise these cognitive domains with appropriate tools [17,26]. On the other hand, the strong correlation we found between SMAC and MSISQ-19 suggests that a negative self-perception of cognitive performance may interfere with sex life as much as the objective cognitive decline detected with neuropsychological tests.

The literature reports depression as a frequent condition in Multiple Sclerosis and is strongly associated with sexual difficulties [11,38,41]. Carotenuto et al. [17] suggest that some depressive symptoms, such as fear of being less attractive, fear of isolation, fear of being left alone, fear of being sexually rejected and issues in communicating can unpleasantly influence sexual activity. In line with these studies, our results showed that depression symptoms play a key role in the relationship between sexual and executive dysfunctions. It is worth noting, indeed, that both executive dysfunction and depression may be due to pre-frontal circuit impairment. High depressive symptoms, indeed, worsened sex life perception, regardless of objective executive performance. Depression is indeed related to reduced self-esteem and perception of failure, but also fatigue and muscular weakness or side effects of medication, and it is often reported in chronic pathologies such as MS [42]. Thus, in this study, depression is confirmed to contribute to the pathogenesis of SD in MS, in addition to specific executive cognitive deficits.

Notwithstanding the value of our results, the study is not without some limitations. Our sample was fairly small and included mostly Relapsing-Remitting patients. The study design would be improved by including a control group of persons with other progressive pathologies or healthy individuals in order to describe possible differences across groups. Pharmacological treatment was not considered as a possible moderating factor due to the limited number of people taking each different type of drug, including first-line (e.g., Interferon, Glatiramer Acetate, Teriflunomide) and second-line medication (e.g., Natalizumab, Fingolimod, Alemtuzumab). Furthermore, no neuroimaging data were available to explore brain damage possibly related to primary sexual dysfunction and cognitive deficits.

Moreover, the current scarcity of existing literature on this subject means caution is required when generalising the results of our research. Further studies are needed to clearly delineate the nature of this interaction, also taking into account how the complex interplay between physiopathological alterations [43] and socioeconomic factors [44] could impact both cognitive and sexual functioning [45]. However, our findings may certainly highlight that executive functions, from a cognitive point of view, and depression, from a psychological point of view, are undoubtedly and intrinsically connected with the sex life of persons with Multiple Sclerosis.

## Figures and Tables

**Table 1 behavsci-13-00369-t001:** Descriptive statistics of participants’ personal data (Age and Education), clinical data (EDSS and Disease duration), and scores at neuropsychological tests and self-report questionnaires. The first row of each variable refers to the whole sample, while the other two refer to males and female separately.

	Mean	SD	Median	Min	Max	Kurtosis	Skewness	Q1	Q3
Age	39.8	11.2	39	19	62	−0.9	0	31.5	47.5
M	34.3	11.8	33	20	62	−0.3	0.7	24	38
F	42.2	10.3	42.5	19	59	−0.7	−0.2	35.2	49
Education	13	3.6	13	5	19	−1	−0.1	10.5	16
M	13.6	3.7	13	8	19	−1.4	−0.2	11	17
F	12.7	3.5	13	5	18	−1	−0.2	10	16
Disease duration	11.2	9.2	10	1	47	3	1.5	4.5	15.5
M	7.8	6	6	1	21	−0.7	0.7	3	11
F	12.7	10	11	1	47	2	1.4	6.2	16.7
EDSS	2.1	1.4	1.5	0	6.5	1.9	1.6	1	2.2
M	2	1.6	1.5	0	6	1.3	1.5	1	2
F	2.1	1.4	1.5	1	6.5	1.8	1.5	1	2.4
MSISQ-19	35.1	15.1	30	19	87	0.6	1	22	46.5
M	31.2	12.6	28	19	63	0	1	21	36
F	36.9	15.9	32.5	19	87	0.4	0.8	24.5	48.5
Primary	9.9	4.9	9	5	24	−0.2	0.9	6	13
M	8.24	3.9	7	5	17	0.1	1.2	5	10
F	10.6	5.2	9	5	24	−0.5	0.7	6	14
Secondary	15.7	6.3	15	9	39	1.5	1.1	10	20
M	14.2	5.3	11	9	25	−0.7	0.8	10	17
F	16.3	6.6	15	9	39	1.4	1.1	11.2	20
Tertiary	9.6	5.1	8	5	24	−0.2	0.9	5	14
M	8.8	4.4	8	5	21	0.9	1.23	5	11
F	9.9	5.4	8	5	24	−0.6	0.8	5	14
SDMT	55	14.3	52	31	109	1.6	0.9	43	65.5
M	57.1	16.9	54	38	109	2.6	1.5	46	63
F	54.1	13.2	50	31	78	−1.3	0.2	43	65.7
LTS-G	43.9	13.2	43	11	67	−0.7	−0.1	35.5	54.5
M	40.8	12.6	42	11	61	−0.1	−0.3	35	48
F	45.3	13.3	45	22	67	−1.1	−0.1	36.7	56
CLTR-G	37.9	14.4	36	8	67	−0.6	0.1	29	49.5
M	35.1	11.6	35	11	61	0	0.2	29	40
F	39.2	15.5	38	8	67	−0.9	0	29.2	52.7
SRT-D	8.7	2.5	9	1	12	0.4	−0.7	7	11
M	7.9	2.3	8	1	11	−0.1	−0.9	7	10
F	9	2.2	9	5	12	−1.1	−0.2	7.2	11
D-KEFS-FSC	10.4	2.2	10	7	16	−0.2	0.6	9	12
M	10.9	1.9	11	9	16	0.8	1	9	12
F	10.2	2.4	10	7	15	−0.6	0.6	8.2	11
D-KEFS-FSD	40.1	9.1	40	25	64	0.1	0.7	34	44
M	41.8	8.1	40	30	64	1	1	36	44
F	39.4	9.5	38	25	60	−0.3	0.7	32	44
D-KEFS-SR	42.2	10.4	43	20	64	−0.4	−0.1	36	48
M	45.9	7.7	44	32	64	−0.2	0.4	42	50
F	40.5	11	40	20	64	−0.6	0.1	33	48
Stroop	58.7	21	55	34	149	8.4	2.6	48.4	63.5
M	56.9	24.4	51.1	34	143	6.3	2.5	48.9	59.2
F	59.5	19.6	56.4	37.8	149	9	2.3	48.5	66.5
BDI-II	9.1	8.1	7	0	34	0	0.8	2.5	14.5
M	5.2	4.5	5	0	16	−0.4	0.7	1	8
F	10.9	8.8	8.5	0	34	−0.6	0.5	3.2	19
SMAC	27.9	18.3	24	2	71	−0.4	0.7	13	36.5
M	20.9	12.8	21	2	54	0.3	0.7	12	27
F	31	19.6	27.5	4	71	−0.9	0.48	13.5	42.5

Note. SD = Standard Deviation; M = Males; F = Females; Q1 = First quartile; Q3 = Third quartile; EDSS = Expanded Disability Status Scale (Kurtzke, 1983); MSISQ-19 = Multiple Sclerosis Intimacy and Sexuality Questionnaire (Sanders et al., 2000; Italian version in Carotenuto et al., 2020); SMAC = Sclerosi Multipla Autopercezione Cognitiva (tr. Self-Perception of cognition in Multiple Sclerosis; Riccardi et al., 2021); SDMT = Symbol Digit Modalities Test (from BRB-NT battery, Rao et al., 1990; Italian version and Italian normative values in Amato et al., 2006); LTS g = Long Term Storage; CLTR g = Consistent Long Term Retrieval; SRT-D = Selective Reminding Test–Delay (from BRB-NT battery, Rao et al., 1990; Italian version and Italian normative values in Amato et al., 2006); D-KEFS-FSC = Delis–Kaplan Executive Function System Sorting Test—Free Sort Categorization (Mattioli et al., 2014); D-KEFS-FSD = Delis–Kaplan Executive Function System Sorting Test—Free Sort Description (Mattioli et al., 2014); D-KEFS-SR = Delis–Kaplan Executive Function System Sorting Test—Sort Recognition (Mattioli et al., 2014); BDI-II = Beck Depression Inventory-II (Beck et al., 1996; Italian version in Ghisi et al., 2006).

**Table 2 behavsci-13-00369-t002:** Regression models with the global score of the Multiple Sclerosis Intimacy and Sexuality Questionnaire-19 (MSISQ-19) as the dependent variable. Model Coefficients and Model Fit values are reported for each model.

Predictors	Model Coefficients		Model Fit		
	Beta	*p*	Adj. R^2^	F-Test	*p*
(intercept)	35.92	0.011			
Age	0.17	0.402			
Education	−0.58	0.361	0.01	1.35	0.268

(intercept)	30.14	<0.001 ***			
EDSS	2.7	0.087			
Disease Duration	−0.05	0.830	0.02	1.69	0.193

(intercept)	68.66	<0.001 ***			
D-KEFS-FSC	−3.22	<0.001 ***	0.217	15.94	<0.001 ***

(intercept)	41.26	<0.001 ***			
D-KEFS-FSC	−1.66	0.01 *			
BDI	1.22	<0.001 ***	0.6	40.58	<0.001 ***

(intercept)	47.34	<0.001 ***			
D-KEFS-FSC	−2.28	0.002 **			
SMAC	0.41	<0.001 ***	0.44	22.27	<0.001 ***

(intercept)	65.25	<0.001 ***			
D-KEFS-FSD	−0.68	0.001 *	0.15	10.64	0.001 **

(intercept)	38.83	<0.001 ***			
D-KEFS-FSD	−0.38	0.013 *			
BDI	1.27	<0.001 ***	0.59	39.92	<0.001 ***

(intercept)	41.93	<0.001 ***			
D-KEFS-FSD	−0.47	0.01 **			
SMAC	0.43	<0.001 ***	0.41	19.81	<0.001 ***

(intercept)	54.17	<0.001 ***			
D-KEFS-SR	−0.45	0.021 *	0.08	5.67	0.08

(intercept)	30.59	<0.001 ***			
D-KEFS-SR	−0.18	0.199			
BDI	1.23	<0.001 ***	0.55	34.48	<0.001 ***

(intercept)	33.95	<0.001 ***			
D-KEFS-SR	−0.27	0.09			
SMAC	0.46	<0.001 ***	0.36	16.55	<0.001 ***

Note. EDSS = Expanded Disability Status Scale (Kurtzke, 1983); D-KEFS-FSC = Delis–Kaplan Executive Function System Sorting Test-Free Sort Categorization (Mattioli et al., 2014); D-KEFS-FSD = Delis–Kaplan Executive Function System Sorting Test-Free Sort Description (Mattioli et al., 2014); D-KEFS-SR = Delis–Kaplan Executive Function System Sorting Test- Sort Recognition (Mattioli et al., 2014); BDI-II = Beck Depression Inventory-II (Beck et al., 1996; Italian version in Ghisi et al., 2006); SMAC = Sclerosi Multipla Autopercezione Cognitiva (tr. Self-Perception of cognition in Multiple Sclerosis; Riccardi et al., 2021). * *p* < 0.05; ** *p* < 0.01; *** *p* < 0.001.

## Data Availability

The data presented in this study are available on request from the corresponding author under a reasonable request.

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
