# Peer review of "Sexual Dysfunction in Multiple Sclerosis: The Role of Executive Function"

_behavsci, 2023, doi:10.3390/bs13050369_

Round 1
Reviewer 1 Report
The authors aimed to explore the relationship between sexual dysfunction, cognitive impairment and depression in people with MS. The manuscript is quite well written and addresses issues relevant to optimizing MS care. I have some comments posted below.
Abstract
Please provide the number of patients with RRMS and PPMS for more precise clinical characteristics of the study group.
Introduction
The aim of the study contained in the introduction is presented quite intricately and is not consistent with that given in the abstract and discussion section.
Methods
Some sentences, e.g. ,,In our sample, EDSS score ranged between 0 and 6.5 (M=2.06; SD=1.45). Disease duration estimated from the onset of the first symptoms ranged between 1 and 47 years (M=11.2; SD=9.24)” should be moved to the Results section.
I am surprised why the authors decided to include 1 patient with PPMS in the study cohort as it is 98% representative of RRMS.
Results
Basic clinical information is missing, i.e. the number of patients using disease-modifying treatments.
Limitations
There was no data about relationship between MRI findings with cognitive decline and sexual dysfunction.
The authors should add conclusions at the end of the discussion section, trying to include directions for future research taking into account the new perspective on the potential impact of cognitive and socioeconomic functions on different functional domains of MS patients (doi: 10.1111/bpa.12220, doi: 10.26444/aaem/117962, doi: 10.1016/B978-0-444-63247-0.00020-1).
Author Response
The authors aimed to explore the relationship between sexual dysfunction, cognitive impairment and depression in people with MS. The manuscript is quite well written and addresses issues relevant to optimizing MS care. I have some comments posted below.
Abstract
- Please provide the number of patients with RRMS and PPMS for more precise clinical characteristics of the study group.
Thank for this suggestion. However, since there is only one PPMS patient and the clinical distinction it is not relevant for the aim of the study we think it is better to leave it out from the abstract. We explained this in the manuscript.
Introduction
- The aim of the study contained in the introduction is presented quite intricately and is not consistent with that given in the abstract and discussion section.
Thank for this suggestion. We rephrased the sentence to improve clarity.
Methods
- Some sentences, e.g. ,,In our sample, EDSS score ranged between 0 and 6.5 (M=2.06; SD=1.45). Disease duration estimated from the onset of the first symptoms ranged between 1 and 47 years (M=11.2; SD=9.24)” should be moved to the Results section.
Thank for this suggestion. Done.
- I am surprised why the authors decided to include 1 patient with PPMS in the study cohort as it is 98% representative of RRMS.
We thank the Reviewer for the comment. We know that 1/54 is not representative of the clinical epidemiology of MS. However, we decided to include also this unique case of PP due to the fact that for our aim the difference among clinical subtypes of MS was not relevant. We specified this in the manuscript.
Results
- Basic clinical information is missing, i.e. the number of patients using disease-modifying treatments.
Thank you for this important point. Our sample was under first- and second-line medications. However, we did not consider this factor within statistical analyses due to the limited sample size. We added this point in the limitations of the paper.
Limitations
- There was no data about relationship between MRI findings with cognitive decline and sexual dysfunction.
Thank you for the observation, we added that point in the limitation section.
- The authors should add conclusions at the end of the discussion section, trying to include directions for future research taking into account the new perspective on the potential impact of cognitive and socioeconomic functions on different functional domains of MS patients (doi: 10.1111/bpa.12220, doi: 10.26444/aaem/117962, doi: 10.1016/B978-0-444-63247-0.00020-1).
We thank the Reviewer for the paper suggested. We incorporated the findings of these studies in the conclusions also suggesting future directions of research towards a multi-disciplinary perspective.
Reviewer 2 Report
Marinetto and colleagues reported on the associations between sexual dysfunction, executive function and depression in multiple sclerosis (n=55). The manuscript is overall clear and well written. Methods are sound enough considering study objective and design. The topic is interesting, considering how common sexual dysfunction is in MS and how difficult is to address. I only have some minor comments to the authors.
In the introduction, when discussing previous papers on this topic, authors should refer to 10.1007/s10072-020-04873-w as well. This study has already showed the impact of depression on sexual dysfunction in MS, and provides some additional hints I believe would be helpful for this paper introduction and discussion.
I believe “biographical data” would better be “demographics”.
In the abstract “The main result highlights that sexual difficulties are predicted by cognitive deficits” and also in the main body of the manuscript, I would avoid the word “predict” and any reference to causal associations. This is a cross-sectional study and authors can only comment on biological plausibility of associations.
Authors showed the interplay between sexual dysfunction and two prefrontal features (executive function and depression). It is perhaps worth commenting on this in the Discussion. Also, it should be disclosed the absence of MRI data which could have shed light on the anatomical and functional correlates of such association.
Author Response
Rev 2
Marinetto and colleagues reported on the associations between sexual dysfunction, executive function and depression in multiple sclerosis (n=55). The manuscript is overall clear and well written. Methods are sound enough considering study objective and design. The topic is interesting, considering how common sexual dysfunction is in MS and how difficult is to address. I only have some minor comments to the authors.
- In the introduction, when discussing previous papers on this topic, authors should refer to 10.1007/s10072-020-04873-w as well. This study has already showed the impact of depression on sexual dysfunction in MS, and provides some additional hints I believe would be helpful for this paper introduction and discussion.
We thank the Reviewer for the comment. We included this paper in the introduction.
- I believe “biographical data” would better be “demographics”.
Thank the Reviewer for this suggestion: done.
In the abstract “The main result highlights that sexual difficulties are predicted by cognitive deficits” and also in the main body of the manuscript, I would avoid the word “predict” and any reference to causal associations. This is a cross-sectional study and authors can only comment on biological plausibility of associations.
We thank Reviewer for this comment. “Predict” here is used in its technical and statistical meaning in regression analyses without implying a causal relationship between cognitive functioning and sexual dysfunction. However, to avoid any misunderstanding, we changed the sentence in accordance with the Reviewer's comment. Now is certainly clearer.
Authors showed the interplay between sexual dysfunction and two prefrontal features (executive function and depression). It is perhaps worth commenting on this in the Discussion.
Thanks for this comment, we indeed added a sentence in the Discussion mentioning this.
Also, it should be disclosed the absence of MRI data which could have shed light on the anatomical and functional correlates of such association.
Thanks for this comment, we indeed added a sentence to clarify this.
Round 2
Reviewer 1 Report
The authors addressed all the comments raised. I have no additional comments.